# Genome Size in the *Arenaria ciliata* Species Complex (Caryophyllaceae), with Special Focus on Northern Europe and the Arctic

**DOI:** 10.3390/plants13050635

**Published:** 2024-02-26

**Authors:** Gregor Kozlowski, Yann Fragnière, Benoît Clément, Olivier Gilg, Benoît Sittler, Johannes Lang, Pernille Bronken Eidesen, Simone I. Lang, Pawel Wasowicz, Conor Meade

**Affiliations:** 1Department of Biology and Botanical Garden, University of Fribourg, Chemin du Musée 10, 1700 Fribourg, Switzerland; yann.fragniere@unifr.ch (Y.F.); benoit.clement@unifr.ch (B.C.); 2Natural History Museum Fribourg, Chemin du Musée 6, 1700 Fribourg, Switzerland; 3Eastern China Conservation Centre for Wild Endangered Plant Resources, Shanghai Chenshan Botanical Garden, 3888 Chenhua Road, Songjiang, Shanghai 201602, China; 4UMR 6249 Chrono-Environment, CNRS, Université de Bourgogne Franche-Comté, 25000 Besançon, France; olivier.gilg@gmail.com; 5Groupe de Recherche en Ecologie Arctique (GREA), 16 rue de Vernot, 21440 Francheville, France; benoit.sittler@nature.uni-freiburg.de (B.S.); johannes.lang@vetmed.uni-giessen.de (J.L.); 6Nature Conservation and Landscape Ecology, University of Freiburg, Tannenbacherstrasse 4, 79106 Freiburg im Breisgau, Germany; 7Arbeitsgruppe Wildtierforschung, Justus-Liebig-University Giessen, Frankfurter Strasse 114, 35392 Giessen, Germany; 8Department of Biosciences, University of Oslo, 0316 Oslo, Norway; p.b.eidesen@ibv.uio.no; 9Department of Arctic Biology, The University Centre in Svalbard, P.O. Box 156, 9171 Longyearbyen, Norway; simonel@unis.no; 10Icelandic Institute of Natural History, Borgum við Norðurslóð, 600 Akureyri, Iceland; pawel@ni.is; 11Molecular Ecology & Biogeography Laboratory, Biology Department, Maynooth University, W23 F2H6 Maynooth, Ireland; conor.meade@mu.ie

**Keywords:** arctic-alpine plants, *Arenaria norvegica*, *Arenaria gothica*, *Arenaria pseudofrigida*, flow cytometry, ploidy

## Abstract

The main aim of the present study has been the completion of genome size data for the diverse arctic-alpine *A. ciliata* species complex, with special focus on the unexplored arctic taxon *A. pseudofrigida*, the north-European *A. norvegica,* and *A. gothica* from Gotland (Sweden). Altogether, 46 individuals of these three Nordic taxa have been sampled from seven different regions and their genome size estimated using flow cytometry. Three other alpine taxa in the *A. ciliata* complex (*A. multicaulis*, *A. ciliata* subsp. *ciliata*, and *A. ciliata* subsp. *bernensis*) were also collected and analyzed for standardization purposes, comprising 20 individuals from six regions. A mean 2c value of 1.65 pg of DNA was recorded for *A. pseudofrigida*, 2.80 pg for *A. norvegica*, and 4.14 pg for *A. gothica,* as against the reconfirmed 2c value of 1.63 pg DNA for the type taxon *A. ciliata* subsp. *ciliata*. Our results presenting the first estimations of genome sizes for the newly sampled taxa, corroborate ploidy levels described in the available literature, with *A. pseudofrigida* being tetraploid (2n = 4x = 40), *A. norvegica* possessing predominantly 2n = 8x = 80, and *A. gothica* with 2n = 10x = 100. The present study also reconfirms genome size and ploidy level estimations published previously for the alpine members of this species complex. Reflecting a likely complex recent biogeographic history, the *A. ciliata* species group comprises a polyploid arctic-alpine species complex characterized by reticulate evolution, polyploidizations and hybridizations, probably associated with rapid latitudinal and altitudinal migrations in the Pleistocene–Holocene period.

## 1. Introduction

Closely related but disjunct arctic–alpine and boreo-montane taxa offer an excellent model system for the study of the influence of polyploidy on evolution and biogeography of plants, especially in the context of historical climate oscillation events [1,2]. These taxa provide a unique opportunity to explore how polyploidy has shaped their evolutionary trajectories and distribution patterns across changing climatic landscapes over time. Investigating their genetic diversity and adaptation strategies can deepen our understanding of how plants respond to environmental shifts and inform conservation efforts in rapidly changing ecosystems.

A classic example of such a group is the *Arenaria ciliata* L. species complex (Caryophyllaceae) comprising six herbaceous taxa with very similar morphology and ecology but with divergent arctic–alpine distribution ranges and ploidy levels [3,4,5,6]. In a recent study of this group, Kozlowski et al. [7] used flow-cytometry and genome size estimations to examine ploidy levels in populations across the Alps and Jura Mountains of central Europe, where four taxa occur in relatively close proximity, namely *A. ciliata* subsp. *ciliata* L., *A. ciliata* subsp. *bernensis* Favarger, *A. gothica* Fr., and *A. multicaulis* L. [4,8], with special focus on *A. ciliata* subsp. *bernensis*, an endemic plant occurring in the Swiss Northern Alps. The focus of the present study is to expand and complete genome size investigation for taxa in Northern Europe and the Arctic, where distribution ranges vary much more widely, and often comprise disjunct populations in reproductively isolated locations.

Genome size is known to vary greatly across organism lineages, including within and between plant groups [9]. This variability arises from a combination of processes leading to, on one hand, duplications and increase in DNA amounts in the genome (particularly polyploidization), and on the other, processes that filter duplications and eliminate DNA (primarily via recombination), in response to both selective pressures and neutral drift events [10]. Correspondingly, nuclear DNA amount and ploidy level are important biodiversity characters and play a significant role in the evolution of land plants, especially with respect to processes of speciation [11,12,13]. As a result, significant recent research efforts have focused on understanding the factors that shape genome size variation [14,15,16].

Recognizing the importance of this information in the context of understanding recent diversification in the *A. ciliata* complex across the Euro-Arctic biogeographic region, we focused our present work on genome size estimation in high-latitude taxa, using the same flow cytometry method as in our previous work [7], for the three unstudied target taxa: *A. pseudofrigida* (Ostenf. & O.C.Dahl) Juz. Ex Schischk. & Knorring and *A. norvegica* Gunn., as well as the Scandinavian populations of *A. gothica* Fr. [17,18,19]. When applied rigorously [20,21], flow-cytometry is established as a benchmark method for estimation of genome sizes and DNA ploidy levels in plants (e.g., [14,22,23,24,25,26]), and provides a replicable protocol for comparison of samples with varying collection, storage, preparation and quantity variables [6]—an important consideration for the assembly of material from remote locations. The present work aims to deliver a targeted synthesis of genomic, biogeographic, and ecological information for three Nordic members of the *A. ciliata* species complex, which have never previously been analyzed collectively.

*Arenaria pseudofrigida* was described as a distinct taxon (*A. ciliata* subsp. *pseudofrigida*) in 1917 from Finnmark (Persfjorden, Vardø) in the far north of Norway, and was raised to a species status by Russian botanists Boris K. Schischkin and Olga E. Knorring in 1936 [27]. Apart of the neighborhood of the *locus classicus* (where the taxon is still well present), there are several isolated populations in Norway and Finland, numerous scattered occurrences in continental European part of the Russian North (e.g., polar Ural), and on several Russian arctic islands (e.g., Novaya Zemlya, Franz Joseph Land). Additionally, the taxon occurs in Svalbard and in Eastern Greenland, where it is relatively frequent (Figure 1) [5,18,28]. In Svalbard, hundreds of occurrences are known from three islands: Spitsbergen (mainly along Isfjorden, Wijdefjorden, and Kongsfjorden), Edge Island, and Prince Charles Foreland [19]. The most northern populations however, are known from north-eastern Greenland, growing as far as 82°30′ N [29].

*Arenaria pseudofrigida* shares a common reproductive strategy typical for the *A. ciliata* species complex. It reproduces sexually via insect pollination (e.g., flies) but is also capable of selfing in the absence of pollinators. Dispersal is nominally quite restricted, as its seeds lack any specialized dispersal mechanisms, relying on wind motion and gravity to dislodge the 0.1–0.5 mm diameter seeds through the neck of the mature fruit capsule, which is indehiscent. Long-distance dispersal (without human influence) is most likely via transport in the digestive tract of birds [17,19].

*Arenaria pseudofrigida* is a specialist of gravelly ground, either along the coast (on raised beaches as on Figure 1D) or on glacial and alluvial deposits in valleys. In Svalbard, it is confined to circumneutral or alkaline soils. The taxon seems to be very tolerant to drought and wind abrasion [19] and is a typical element of the boreal and arctic base-rich scree and block fields (alliance *Arenarion norvegicae* Nordhagen 1935) [30]. The taxon was described as tetraploid, with 2n = 4x = 40 [18,19,31].

*Arenaria norvegica* was described by Johann Ernst Gunnerus at the end of the 18th century, from Laskenstad (isle of Steigen, Northern Norway) in his *Flora Norvegica* [18,32]. The taxon occurs in Iceland (Figure 1) and Norway, where it is quite frequent, as well as possesses scattered and partially extremely small and isolated populations in Sweden, Finland, Shetland Islands, northern Scotland, western Ireland, and northern England [5,17,18]. The English population is sometimes differentiated as a separated subspecies (*A. norvegica* subsp. *anglica* Halliday), endemic to mid-west Yorkshire [17,33]. *Arenaria norvegica* is found on base-rich sandy substrates and in fine scree and on riverside gravels (Figure 1) [5,17]. Similarly to *A. pseudofrigida*, the species is also very resistant to drought and wind, being able to prosper, for example, in the desert of lava, gravel and blown sand in the central plateau of Iceland [34]. According to the literature, it is an octoploid taxon with 2n = 8x = 80 [18,19,31].

*Arenaria gothica*, described from the isle of Gotland (Baltic Sea, Sweden) by Wahlenberg [35], was raised to a species rank by Fries [36]. Later, a similar form was discovered in Switzerland, along the shore of the Lac des Joux [37] and was judged by Grenier [38] to be conspecific with *A. gothica* Fr. In the second half of the 19th century, an additional occurrence was found on the Swedish mainland (Kinnekulle, Västergötland) [39,40]. *Arenaria gothica* has therefore an extremely disjunct distribution (distribution maps in [39] and [40]), and its taxonomic position, putative origin and its glacial relict status is a subject of controversy [33,40,41]. The ecology, particularity of the Swiss populations of *A. gothica*, has been explored in detail in our recent work [7]. In Sweden the taxon occurs on chalk and limestone, mainly in open habitat called *alvar* [40]. Alvars are almost level areas that are only sparsely covered by vegetation of the order *Alysso-Sedetalia* (basiphilous dry grasslands of shallow, skeletal soils), and are restricted to the Baltic islands of Sweden and to Estonia [39,42]. The chromosome number from both disjunct regions was counted as 2n = 10x = 100 [7,40].

The aim of the present study was to complement our earlier work exploring alpine and Jura Mountain taxa of the *A. ciliata* complex [7] and deliver the first evaluation of genome size variation for yet unstudied taxa from Northern Europe and the Arctic. In this way, the main focus of the present work is on *A. pseudofrigida* and *A. norvegica*, as well as on the Swedish populations of *A. gothica* from Gotland. The following specific questions have been addressed: (1) What are the differences in genome size among the three north-European and arctic species of the complex, in comparison with closely related taxa occurring in the Alps and neighboring mountain ranges? (2) Do the obtained results corroborate the ploidy levels of the studied taxa known from the literature? Based on our results, the influence of the ploidy level on the evolutionary and biogeography history of the *A. ciliata* species complex will be discussed.

## 2. Results

The 2c values recorded for *A. pseudofrigida* were varying between 1.54 pg and 1.84 pg of DNA (Figure 2, Appendix A), with a mean 2c value of 1.65 pg (Table 1, standard deviation, SD ± 0.11). The recorded genome size of *A. norvegica* was higher, with 2c values varying between 2.68 pg and 2.89 pg of DNA, with a mean 2c value of 2.80 pg (SD ± 0.02). The third Nordic taxon, *A. gothica*, displayed the highest recorded values, varying between 3.90 pg and 4.41 pg of DNA, with mean 2c value of 4.14 (SD ± 0.26). The higher SD-values could be attributed to two different storage techniques of the plant material, as explained in the Materials and Methods. Plant tissue desiccated with silica gel and stored at room temperature showed higher 2c values (4.33 and 4.41) than duplicate samples stored for a longer time at −20 °C (3.90 and 3.94). These differences are most probably due to technical issues, and not natural variation among populations, as explained, for example, by Sliwinska et al. [20].

*Arenaria ciliata* subsp. *ciliata* and *A. multicaulis*, both showed similar but much lower values, with a mean 2c value of 1.63 (SD ± 0.06) for *A. ciliata* subsp. *ciliata* and a mean 2c value of 1.50 (SD ± 0.06) for *A. multicaulis* (Figure 2, Table 1). The highest 2c values among all the six taxa from the *A. ciliata* complex investigated in this study were recorded for *A. ciliata* subsp. *bernensis,* varying between 6.51 pg and 7.02 pg of DNA, with a mean 2c value of 6.77 pg (SD ± 0.03). The results show that, for all six taxa taken separately, the genome size is very stable, thus indicating an invariant ploidy level of all investigated individuals within a given taxon (Figure 2, Appendix A).

## 3. Discussion

The study identified stable but varying genome sizes across different taxa studied within the *A. ciliata* species complex. It is composed exclusively of polyploid taxa, with no diploids. Polyploid species complexes play an important role in forming local floras [46]. The *A. ciliata* group is one of many polyploid arctic–alpine complexes characterized by reticulate evolution, polyploidizations and hybridizations. Well-studied similar complexes exist elsewhere in the family Caryophyllaceae, for example, in the genera *Cerastium* [47] and *Silene* [48]. Polyploid species complexes are also observed in numerous other well-documented circumpolar, arctic–alpine, and alpine genera and families, for example, in *Draba* (Brassicaceae, [49]), *Primula* (Primulaceae, [50,51]) and *Calamagrostis* (Poaceae, [18,19,52]).

Our present study, focused on north-European and arctic members of the *A. ciliata* species complex, completes genome size estimations initiated in Kozlowski et al. [7]. Although allowing for the potential limitations of genome size estimations using flow cytometry [20,21], this technique facilitates a synthesis of ploidy level variability within the target arctic–alpine plant group. According to our results, the following three taxa are predominantly tetraploid (2n = 4x = 40): the arctic *A. pseudofrigida* and the two alpine taxa *A. multicaulis* and *A. ciliata* subsp. *ciliata*. Higher ploidy is observed in *A. norvegica* (2n = 8x = 80) and *A. gothica* (2n = 10x = 100), and the highest confirmed ploidy level is for the narrow endemic of Western Alps *A. ciliata* subsp. *bernensis* (2n = 20x = 200). These estimates corroborate chromosome counts available in the published literature [18,19,33,40,41,43,44,45]. However, for some taxa of the present study, greater variability in ploidy has been documented in previous work, for example, for *A. ciliata* subsp. *ciliata* (2n = 40, 80, 120, 160, 200), but also for *A. norvegica* (2n = 60, 80) [6,8,44]. Abukrees et al. [6] showed that such atypical chromosome counts could be detected in 12.5% of investigated individuals for *A. norvegica* and in as much as 37% of investigated individuals for *A. ciliata* subsp. *ciliata*. In contrast, the available literature and our results demonstrate a very stable ploidy level in the remaining four taxa, namely *A. pseudofrigida* and *A. multicaulis* (both tetraplid), *A. gothica* (decaploid) and *A. ciliata* subsp. *bernensis* (dodecaploid), and was this across the whole investigated distribution area.

Our results thus confirm that there are no extant diploid taxa within the *A. ciliata* species complex. The group comprises an example of a so-called *mature polyploid complex* in the sense of Stebbins [53]. This is not an exception among arctic–alpine disjunct taxa. A similar polyploidy pattern can be observed, for example, in the *Cerastium alpinum/arcticum* species complex [47] and in the *Calamagrostis stricta/neglecta* group [19,52], where the lowest chromosome number known is tetraploid (for *Calamagrostis*) or even octoploid (*Cerastium*). In addition, the *A. ciliata* species complex confirms the conclusion of Brochmann et al. [1], that almost 90% of arctic specialist plant taxa growing in regions that were heavily glaciated during the last ice age, are polyploids.

Polyploidy is thought to infer fitness advantages allowing plants to adapt better to more extreme climatic conditions [54], and in this scenario, it may not be surprising that arctic and northern-latitude habitats include some of the most polyploid-rich floras [1]. In a related inference, it has been postulated that there is a general increasing gradient of polyploidy with latitude (and altitude) [55]. This latter hypothesis is now refuted. Stebbins [56] concluded, for example—by comparing polyploidy along the Pacific Coasts of North America—that the highest frequency of polyploids occurs at between 52° and 54° north, declining to the north and south. Additionally, the majority of typical arctic polyploid complexes (for example, in the genera *Calamagrostis, Campanula, Chamaenerion, Salix*, and *Saxifraga*), reveal the presence of diploids at their northern distributional limits. Similarly, the *A. ciliata* species complex explored in our study does not show a specific latitudinal ploidy gradient, with the lowest chromosome numbers in high arctic *A. pseudofrigida* as well as in the majority of high alpine populations of *A. ciliata* subsp. *ciliata* and *A. multicaulis*. It is the case, however, that the highest ploidy types are only found at high elevation, near the geographic center of diversity for the species complex.

The most plausible hypothesis explaining the high polyploid frequency in arctic–alpine plants, is the so-called *secondary contact hypothesis* [56]. According to this hypothesis, “polyploidy, accompanied by hybridization, is instrumental chiefly for rapid adaptation to new ecological conditions, that become available relatively suddenly” [56]. Migration at the beginning and the end of warm and cold periods is likely to be an important driver of ploidy diversification, particularly for recently evolved polyploids [57], with multiple recurrence of secondary contacts between previously separated closely related taxa [56]. Considering the historical and current geographic ranges of *A. ciliata* complex taxa, including the proliferation of extant populations in recently deglaciated regions of Europe and the Arctic, it is highly likely that secondary contacts played an important role in evolution of high polyploidy in the *A. ciliata* species complex. Our study corroborates conclusions of Abukrees et al. [6], stating that the *A. ciliata* complex arose from a reduced ploidy ancestral stock (2n = 40), probably in the Alps, which after the latitudinal and altitudinal migration gave rise to several polyploidization events. *Arenaria ciliata* subsp. *bernensis* (2n = 200), for example, is proposed to be an allopolyploid neo-endemic taxon resulting from hybridization between different related taxa due to rapid migration events after the last glaciation period (probably, in this case, of *A. multicaulis* and A. *ciliata* subsp. *ciliata*) [4,7,45,58]. The genome size and, thus, also the ploidy level, is stable across the whole distribution area of this taxon. The present study delivers additional support for the taxonomic distinctiveness of the high alpine endemic *A. ciliata* subsp. *bernensis*, which strongly aligns with other differences in morphology, phylogeny, phenology, ecology, and plant communities, described previously. In affirming these differences, further support now exists to re-consider the species status of this taxon. A similar allopolyploid origin after the last glaciation was proposed for *A. gothica* (2n = 100), with *A. norvegica* (2n = 80) and *A. serpyllifolia* subsp. *leptoclados* (2n = 20) as potential parents [33], or between *A. multicaulis* (2n = 40) and *A. serpyllifolia* subsp. *leptoclados* [41]. In this way, ploidy and genome size data from present study confirm that taxonomic and distributional differences among northern and arctic taxa of the *A. ciliata* complex are likely reflecting a discrete genetic origin and migratory history in each case.

## 4. Materials and Methods

### 4.1. Sampling of Plant Material

Most of the individuals of the *A. ciliata* species complex were recently collected in the field, explicitly for this study. *Arenaria pseudofrigida* was collected in July 2023 (21 individuals from 2 regions in Northeast Greenland, Figure 3, Appendix A). Plant material of *A. norvegica* was sampled in June 2023, mainly in Iceland (21 individuals from 4 populations in western Iceland, Figure 3, Appendix A).

Additionally, one individual from Scotland (UK) was included in the analysis from collections of one of the authors (C. Meade, Abukrees et al. [6]). Similarly, four individuals of *A. gothica* collected previously from Gotland in Sweden were also included in the present study. In addition, three alpine taxa of the *A. ciliata* species complex were collected in August and September 2023 in the Swiss Alps: 11 individuals of *A. ciliata* subsp. *bernensis* from 3 summit areas, 4 individuals of *A. ciliata* subsp. *ciliata* from 2 populations, and 5 individuals from one population of *A. multicaulis* (Figure 3, Appendix A). The voucher specimens are stored in the herbarium of the Natural History Museum Fribourg (NHMF), Switzerland. None of these alpine populations and individuals were included in our previous study [7]. Plant material (small portion of flowering stem with flowers) was silica dried and kept for ca. 4 weeks in plastic bags prior to flow cytometry analyses. An exception was the sole sample from Scotland (*A. norvegica*) and two samples of *A. gothica* from Sweden which were desiccated and then stored at −20 °C prior to analyses. The whole plant material was sent for analysis to the Plant Cytometry Services (Didam, The Netherlands, www.plantcytometry.nl, accessed on 15 December 2023), a biological research company specialized for ploidy and genome size analysis.

### 4.2. Flow Cytometry Analysis

Approximately 1 cm^2^ of leaves of the *Arenaria* samples were mixed with 1 cm^2^ of fresh leaves of the standard plants (*Allium schoenoprasum*, genome size 2c = 15.03 pg). This was chopped with a sharp razor blade to release the nuclei in 100 µL of CyStain nuclei extraction buffer (Sysmex, Norderstedt, Germany, https://eu.sysmex-flowcytometry.com, accessed on 25 December 2023). The obtained suspension was then sieved through a 40 µm filter, and 1.5 mL of CyStain PI (propidium iodide) absolute P staining buffer was added. After one hour, the fluorescence of nuclei in the suspension was measured using a Sysmex ploidy analyzer (Sysmex, Norderstedt, Germany). Each individual was analyzed once. The number of nuclei measured for each sample was large enough in order to determine the ploidy and ranged between 100 and 1000 nuclei per sample (Appendix A). The use of higher nuclei numbers would influence the genome size estimations by 1–2%. Ploidy level was estimated based on comparison against all previously published 2c values and chromosome counts (e.g., [6,7,8,18,19,33,40,41,43,44,45]), as well as on our own chromosome counting in the *A. ciliata* species complex [7].

## Figures and Tables

**Figure 1 plants-13-00635-f001:**
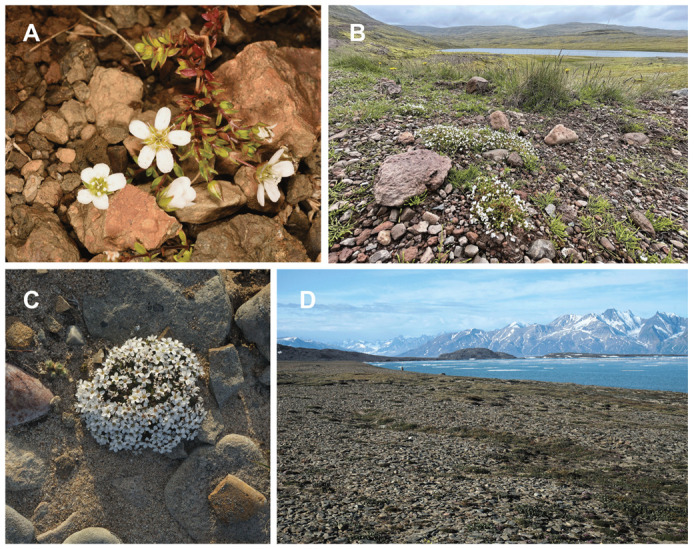
Habitats and morphology of the two most Nordic members of the *Arenaria ciliata* species complex. (**A**,**B**): *Arenaria norvegica*. (**A**) Reykholahreppur, Iceland; (**B**) Latrabjarg, Iceland. (**C**,**D**): *Arenaria pseudofrigida*, Traill Island, Karupelv, Greenland. Photos: (**A**,**B**): Gregor Kozlowski; (**C**,**D**): Sven Büchner.

**Figure 2 plants-13-00635-f002:**
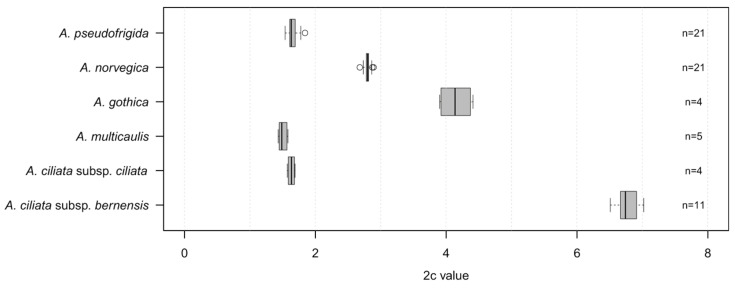
Variation in genome size of four taxa in the *Arenaria ciliata* species complex analyzed in this study (2c values are given in pg of DNA). Box plots showing the quartiles, the 5th and 95th percentiles (whiskers) and the outliers. The number of analyzed individuals is indicated on the right of the plot.

**Figure 3 plants-13-00635-f003:**
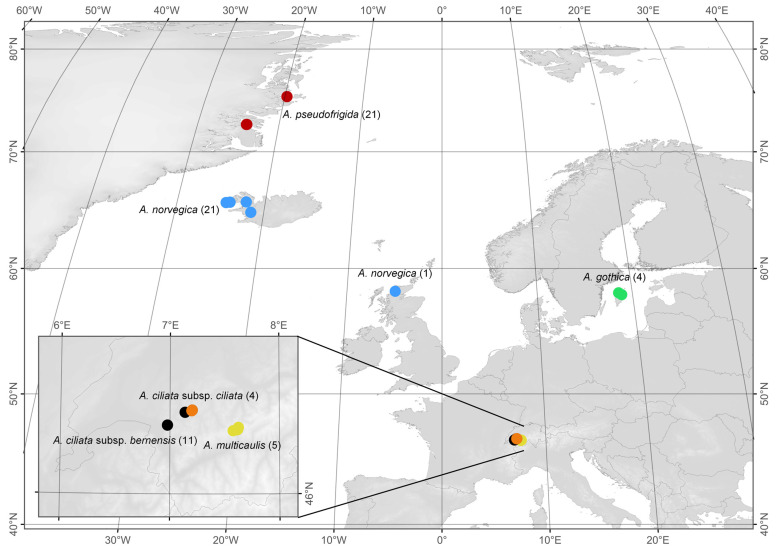
Geographic position of collected sites with *Arenaria ciliata* species complex, with the numbers of individuals sampled in the given area in parentheses. Red: *A. pseudofrigida*; blue: *A. norvegica*; green: *A. gothica*; yellow: *A. multicaulis*; orange: *A. ciliata* subsp. *ciliata*; black: *A. ciliata* subsp. *bernensis*.

**Table 1 plants-13-00635-t001:** Estimated ploidy level and genome size (mean ± standard deviation) in *Arenaria ciliata* species complex in the Arctic, Northern Europe, and the Alps. The genome size (2c values) is given in pg of DNA. The estimated ploidy level is based on comparison against all previously published 2c values and chromosome counts (e.g., [6,7,8,18,19,33,40,41,43,44,45]).

Taxon	2c Nuclear DNA Amount (pg DNA), Mean (± SD)	Estimated Ploidy Level
Nordic taxa:		
*Arenaria pseudofrigida*	1.65 (±0.11)	2n = 4x = 40
*Arenaria norvegica*	2.80 (±0.02)	2n = 8x = 80
*Arenaria gothica* (Gotland)	4.14 (±0.26)	2n = 10x = 100
Alpine taxa:		
*Arenaria multicaulis*	1.50 (±0.06)	2n = 4x = 40
*Arenaria ciliata* subsp. *ciliata*	1.63 (±0.06)	2n = 4x = 40
*Arenaria ciliata* subsp. *bernensis*	6.77 (±0.03)	2n = 20x = 200

## Data Availability

Data are contained within the article and supplementary materials.

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
