# Peer review of "Genome Size in the Arenaria ciliata Species Complex (Caryophyllaceae), with Special Focus on Northern Europe and the Arctic"

_plants, 2024, doi:10.3390/plants13050635_

Round 1

Reviewer 1 Report

Comments and Suggestions for Authors

I have reviewed the manuscript concerning the genome size measurement of Arenaria ciliata species complex in arctic alpine regions. In 2022, the authors published a paper focusing on four species of the complex and they collected data for two additional species not previously studied, totaling six species in this analysis.

The methodology and experimental procedures employed in this study are relatively simple; however, the authors have effectively ensured data reliability by obtaining an adequate number of specimens from various regions and appropriately applying statistical analyses to the results.

In particular, the authors have logically developed their arguments by integrating results on chromosome numbers from existing literature, associating them with genome size within the taxonomic group, levels of polyploidy, and ecological adaptations of the species.

I would like to offer the following comments for the authors' consideration:

1. Lines 169-173

The authors mentioned discrepancies in 2C values based on storage methods after sample collection. It would be beneficial for readers if the authors could provide evidence or comparative cases to help determine whether these differences are due to technical issues or natural variations among specimens from different regions.

2. Fig. 3: Assigning different colors to dots corresponding to each species would enhance the clarity, aiding in visual interpretation.

3. Line 294: "15,03 pg" should be corrected to "15.03 pg" for consistency and clarity, using a period instead of a comma.

Author Response

Answers to reviewers

Our answers are given after each point/question.

Reviewer 1

Comments and Suggestions for Authors

I have reviewed the manuscript concerning the genome size measurement of Arenaria ciliata species complex in arctic alpine regions. In 2022, the authors published a paper focusing on four species of the complex and they collected data for two additional species not previously studied, totaling six species in this analysis.

The methodology and experimental procedures employed in this study are relatively simple; however, the authors have effectively ensured data reliability by obtaining an adequate number of specimens from various regions and appropriately applying statistical analyses to the results.

In particular, the authors have logically developed their arguments by integrating results on chromosome numbers from existing literature, associating them with genome size within the taxonomic group, levels of polyploidy, and ecological adaptations of the species.

Answer: Thank you for your positive reaction to our manuscript.

I would like to offer the following comments for the authors' consideration:

  1. Lines 169-173

The authors mentioned discrepancies in 2C values based on storage methods after sample collection. It would be beneficial for readers if the authors could provide evidence or comparative cases to help determine whether these differences are due to technical issues or natural variations among specimens from different regions.

Answer: Such differences are due to technical issues, as explained by Sliwinska et al. (2021). Short explicative text is now added along with the new reference.

  1. Fig. 3: Assigning different colors to dots corresponding to each species would enhance the clarity, aiding in visual interpretation.

Answer: Done, new version of the Figure is now included.

  1. Line 294: "15,03 pg" should be corrected to "15.03 pg" for consistency and clarity, using a period instead of a comma.

Answer: Corrected.

Reviewer 2 Report

Comments and Suggestions for Authors

This study reports on completion of a dataset for nuclear genome size in Arenaria ciliata complex by estimating nuclear DNA content in A. ciliata group taxa from Norther Europe and Arctic. Unfortunately, the study suffers from serious deficiencies. The authors used flow cytometry to estimate ploidy and genome size. However, the methodology is not described well and important information is missing.  My major concern concerns the estimation of ploidy. In reality, the authors simply estimated ploidy based on nuclear DNA content without a direct comparison with accessions where ploidy is known after chromosome counting (c.f.. Sliwinska et al.,  Cytometry A. 2022 Sep;101(9):749-781. doi: 10.1002/cyto.a.24499). The simplified way ploidy was estimated in this study should be clearly stipulated throughout the paper and possible risk of errorneous estimation explained.

Specific comments

·        Introduction, Line 93: Please specify the name of the Russian botanist (ideally include a reference to a publication).

·        Results

o   Table 1: I recommend changing the heading of column 2 to “2C Nuclear DNA amount (pg DNA).  I also suggest adding a column reporting 1Cx DNA amounts. Moreover, please, include a comment to column 2 (Estimated ploidy level) to explain how the ploidy was estimated.

o   I suggest including a Table 2 giving a list of all species and individuals that were analyzed, including geographical locations.

o   Please, include a figure with two or more representative histograms of DNA content estimation.

o   Please, specify if herbarium vouches were prepared and if so, where they are stored.

·        Discussion: This part of the chapter discusses extensively the results of ploidy estimation. However, given that ploidy was actually not analyzed in this work, I recommend that this part of the Discussion is condensed.

·        Materials and Methods, Lines 293 - 300:

o   Please explain the choice of DNA reference standards, whose DNA amounts differ dramatically from the sampled plants. Such difference may negatively affect the precision of the estimates.

o   Please correct the spelling: Cysrain = CyStain; Pi = PI

o   Please use decimal points and not commas for DNA amount data.

o   Please, specify how many times was each individual analyzed.

o   Please, report on the coefficient of variation (CV) of DNA peaks on histograms of relative DNA content

Comments on the Quality of English Language

No particular comments.

Author Response

Answers to reviewers

Our answers are given after each point/question.

Reviewer 2

(x) Moderate editing of English language required

Answer: The whole manuscript has been checked by native English speaker and one of the co-authors of the manuscript (Conor Meade). All English improvements are done with the track changes and are visible in the resubmitted manuscript.

Comments and Suggestions for Authors

This study reports on completion of a dataset for nuclear genome size in Arenaria ciliata complex by estimating nuclear DNA content in A. ciliata group taxa from Norther Europe and Arctic. Unfortunately, the study suffers from serious deficiencies. The authors used flow cytometry to estimate ploidy and genome size. However, the methodology is not described well and important information is missing.  My major concern concerns the estimation of ploidy. In reality, the authors simply estimated ploidy based on nuclear DNA content without a direct comparison with accessions where ploidy is known after chromosome counting (c.f.. Sliwinska et al., Cytometry A. 2022 Sep;101(9):749-781. doi: 10.1002/cyto.a.24499).

Answer: Thank you for the valuable remarks and constructive criticism. Concerning the succinct description of the methodology: Indeed, it was described very succinctly in this paper. The main reason was however, that this paper is a finalization/completion of the previous work (Kozlowski et al. 2002) published in the same Special Issue of Plants. In fact, we are making comparisons with ploidy after chromosome counting, as explained in Kozlowski et al. 2022 (counting for A. ciliata subsp. bernensis). We cited new literature as proposed by the reviewer (Sliwinska et al. 2021).

The simplified way ploidy was estimated in this study should be clearly stipulated throughout the paper and possible risk of errorneous estimation explained.

Answer: We were already very prudent with our statements, for example in the Abstract we are writing “The present study reinforces additionally the genome size and ploidy level estimations published previously…”. Nevertheless, an according to the wish of the reviewer, we have weakened our statements also in other places, and have more clearly stipulated throughout the manuscript that our work provides only an additional argumentation and indirect confirmation of ploidy levels. Additionally, we cited new literature (e.g., Sliwinska et al. 2021, Temsch et al. 2021) discussing these issues.

Specific comments

Introduction, Line 93: Please specify the name of the Russian botanist (ideally include a reference to a publication).

Answer: Done (the names of botanist are now added and with new reference).

Results

Table 1: I recommend changing the heading of column 2 to “2C Nuclear DNA amount (pg DNA).  I also suggest adding a column reporting 1Cx DNA amounts. Moreover, please, include a comment to column 2 (Estimated ploidy level) to explain how the ploidy was estimated.

Answer: All points are corrected, with one exception (1C values). According to Sliwinska et al. (2021, reference proposed by the reviewer), the reporting in a publication should always use the same values (either always 1C or always 2C). And that 2C is more biologically meaningful in vascular plants. Thus, we decided to use consistently 2C values in the whole manuscript (and thus also in the Table 1).

I suggest including a Table 2 giving a list of all species and individuals that were analyzed, including geographical locations.

Answer: This information is given in detail in the Table S1, with geographical location, etc. Since this table is very long, we would rather prefer to keep it as a Supplementary Material.

Please, include a figure with two or more representative histograms of DNA content estimation.

Answer: We have added a new figure, giving selected examples of histograms in the Supplementary Material.

Please, specify if herbarium vouches were prepared and if so, where they are stored.

Answer: The following text fragment has been added in the Materials and Methods: The voucher specimens were collected and are stored in the herbarium of the Natural History Museum Fribourg (NHMF), Switzerland.

Discussion: This part of the chapter discusses extensively the results of ploidy estimation. However, given that ploidy was actually not analyzed in this work, I recommend that this part of the Discussion is condensed.

Answer: We would like kindly request the reviewer and editors, to accept our Discussion section as it is. This is one of the most important parts of our manuscript, delivering the synthesis of what is known about this arctic-alpine species complex. This was never published and summarized before in one compact paper. We added an additional sentence at the end of the Discussion in order to explain the importance of the present paper and its conclusions.

Materials and Methods, Lines 293 - 300:

Please explain the choice of DNA reference standards, whose DNA amounts differ dramatically from the sampled plants. Such difference may negatively affect the precision of the estimates.

Answer: Since we knew from our previous study, that Arenaria ciliata species complex shows large range of 2C values (from 1.5 to 7), we have decided to use a standard with relatively high value (Allium shoenoprasum, 2C = 15.03), in order to use the same standard for all taxa of the species complex, and to have a clear differentiation between the taxa. Additionally, when changing the Methods chapter, we have found an error: Clivia miniata was not used in this study. Thus, we eliminated this standard taxon from the text. More generally, and based on other remarks of reviewers, we have rewritten or improved other fragments of Materials and Methods.

Please correct the spelling: Cysrain = CyStain; Pi = PI

Answer: Done.

Please use decimal points and not commas for DNA amount data.

Answer: Done.

Please, specify how many times was each individual analyzed.

Answer: We have added the information in the Methods (each individual was analyzed once). However, for the main taxa of this study (A. pseudofrigida and A. norvegica) more than 20 individuals were used in the analysis, with very low standard deviations for the measurements (+/- SD between 0.02 and 0.11).

Please, report on the coefficient of variation (CV) of DNA peaks on histograms of relative DNA content

Answer: Since the standard deviations of all measurements are very low, we have decided to not report the coefficient of variation for the histograms (which was at average 6-7%). The selected examples of CV are now given in the new Figure S+ in the supplementary material.

Round 2

Reviewer 2 Report

Comments and Suggestions for Authors

The authors modified the text as suggested. However, judging from the histograms of DNA content (Figure S1), very small number of nuclei were analyzed in individual samples. The authors should include the number of nuclei analyzed in each sample in the section Materials and Methods and explain what was the reason for this limitation. It appears that the flow cytometric analysis was done by a service provider. If so, this information should also be included.

Author Response

Reviewer 2 (2nd run):

The authors modified the text as suggested.

Answer: Thank you for your positive remark.

However, judging from the histograms of DNA content (Figure S1), very small number of nuclei were analyzed in individual samples. The authors should include the number of nuclei analyzed in each sample in the section Materials and Methods and explain what was the reason for this limitation.

Answer: The number of nuclei used in each sample is now included in the Table S1 (new version of this Supplementary Material in the attachment). Additionally, we added an additional explanatory text in the Materials and Methods.

It appears that the flow cytometric analysis was done by a service provider. If so, this information should also be included.

Answer: This information is included in the manuscript, at the end of the Chapter 4.1.

Round 3

Reviewer 2 Report

Comments and Suggestions for Authors

The authors modified the text as suggested. Just please replace commas by decimal points in the Table S1.